# Contribution of CYP2D6 Functional Activity to Oxycodone Efficacy in Pain Management: Genetic Polymorphisms, Phenoconversion, and Tissue-Selective Metabolism

**DOI:** 10.3390/pharmaceutics13091466

**Published:** 2021-09-14

**Authors:** Malavika Deodhar, Jacques Turgeon, Veronique Michaud

**Affiliations:** 1Precision Pharmacotherapy Research and Development Institute, Tabula Rasa HealthCare, Orlando, FL 32827, USA; mdeodhar@trhc.com (M.D.); jturgeon@trhc.com (J.T.); 2Faculty of Pharmacy, Université de Montréal, Montréal, QC H3T 1J4, Canada

**Keywords:** oxycodone, pharmacogenetics, *CYP2D6*, *OPRM1*, *COMT*, efficacy, oxymorphone

## Abstract

Oxycodone is a widely used opioid for the management of chronic pain. Analgesic effects observed following the administration of oxycodone are mediated mostly by agonistic effects on the μ-opioid receptor. Wide inter-subject variability observed in oxycodone efficacy could be explained by polymorphisms in the gene coding for the μ-opioid receptor (*OPRM1*). In humans, oxycodone is converted into several metabolites, particularly into oxymorphone, an active metabolite with potent μ-opioid receptor agonist activity. The CYP2D6 enzyme is principally responsible for the conversion of oxycodone to oxymorphone. The *CYP2D6* gene is highly polymorphic with encoded protein activities, ranging from non-functioning to high-functioning enzymes. Several pharmacogenetic studies have shown the importance of CYP2D6-mediated conversion of oxycodone to oxymorphone for analgesic efficacy. Pharmacogenetic testing could optimize oxycodone therapy and help achieve adequate pain control, avoiding harmful side effects. However, the most recent Clinical Pharmacogenetics Implementation Consortium guidelines fell short of recommending pharmacogenomic testing for oxycodone treatment. In this review, we (1) analyze pharmacogenomic and drug-interaction studies to delineate the association between CYP2D6 activity and oxycodone efficacy, (2) review evidence from CYP3A4 drug-interaction studies to untangle the nature of oxycodone metabolism and its efficacy, (3) report on the current knowledge linking the efficacy of oxycodone to *OPRM1* variants, and (4) discuss the potential role of CYP2D6 brain expression on the local formation of oxymorphone. In conclusion, we opine that pharmacogenetic testing, especially for *CYP2D6* with considerations of phenoconversion due to concomitant drug administration, should be appraised to improve oxycodone efficacy.

## 1. Introduction

Drug overdose deaths involving prescription opioids (natural and semi-synthetic opioids and methadone) rose from 3442 in 1999 to 14,139 in 2019, reaching a peak of 17,209 deaths in 2017 [1]. Oxycodone is a semi-synthetic opioid used for pain management, accounting for approximately 17 million prescriptions in 2018 [2]. In 2016–2017, oxycodone prescriptions represented approximately 18% of all pain management prescriptions following an emergency room visit in the United States [3]. Recently, the role of genetic polymorphisms associated with proteins involved in the pharmacokinetics and pharmacodynamics of oxycodone was reviewed by the Clinical Pharmacogenetics Implementation Consortium (CPIC), which concluded that current evidence was insufficient to make firm clinical recommendations—in the form of a clinical guideline—regarding genetic testing for oxycodone dose adjustment [4]. Some study results have clearly demonstrated and supported the value of genetic testing, though others have not found clinical value in genetic testing to improve pain management with oxycodone. This is especially the case for polymorphisms associated with *CYP2D6* and the formation of oxymorphone—an active metabolite of oxycodone—and for polymorphisms linked to the opioid receptor mu 1 gene (*OPMR1*) [5].

This review uses the Quality of Evidence per GRADE Criteria for each section presented [6], and evidence supporting the role of relevant pharmacokinetics, pharmacodynamics, and pharmacogenomics information for pain management with oxycodone is discussed. This review also aims to explain why discrepancies exist between results from fundamental research and some clinical studies.

## 2. Pharmacokinetic Considerations

### 2.1. CYP2D6 Activities, Polymorphisms, and Expression (GRADE High Quality ++++)

The cytochrome P450 (CYP450) superfamily is comprised of several heme-containing monooxygenases involved in the phase I metabolism of ~75% of small molecules and is a crucial part of our current pharmacological armamentarium. In humans, this superfamily of enzymes is encoded by 57 genes from 18 families [7]. One gene, *CYP2D6*, codes for an enzyme involved in the metabolism of eicosanoids, drugs, and foreign chemicals [8,9,10,11]. *CYP2D6* is a highly polymorphic gene, and more than 100 variants and subvariants have been identified [12]. CYP2D6 substrates are typically lipophilic bases with an aromatic ring and a nitrogen atom that are protonated at physiological pH. Substrate binding is then followed by oxidation 5–7 Å, apart from the nitrogen moiety [13,14]. CYP2D6 is responsible for the clearance of at least 20% of the compounds in current clinical use, including antiarrhythmics, antidepressants, antipsychotics, β-blockers, and analgesics [9,10,15].

As *CYP2D6* allelic variants influence protein expression and activity, *CYP2D6* polymorphisms affect the functional capability of the enzyme to metabolize CYP2D6 substrate drugs. Allele combinations can produce poor metabolizers (PMs: non-functioning CYP2D6), intermediate metabolizers (IMs: low-functioning CYP2D6), normal metabolizers (NMs: normal functioning CYP2D6), and ultra-rapid metabolizers (UMs: high-functioning CYP2D6, mostly due to gene duplication). In a large genetic testing study of 104,509 deidentified patient samples, it was extrapolated that in the population, an estimated 5.7% of individuals would be PMs, 10.7% would be IMs, and 2.2% would be UMs, with 81% NMs [16]. However, there are differences in *CYP2D6* allele frequencies in distinct racial and ethnic populations. For example, the *CYP2D6*4* allele frequency appears higher in Caucasians, *CYP2D6*10* in East Asians, *CYP2D6*41* and duplication/multiplication of active alleles in those of Middle Eastern descent, *CYP2D6*17* in Black Africans, and *CYP2D6*29* in African Americans [17,18,19].

CYP2D6 is highly and selectively expressed in the human liver. Few tissues express CYP2D6 and recent studies demonstrated that when using an absolute protein assay and LC-MSMS technique, the protein could not be detected in various regions of the human small intestine [20,21]. In 1987, Fonne-Pfister et al. demonstrated the expression of CYP2D6 in the human brain [22]. More than 10 years later, dextromethorphan—a well-known CYP2D6 probe substrate—to dextrorphan metabolism was demonstrated in human brain microsomes [23]. CYP2D6 expression has also been described in various brain regions and cell types including the cerebral cortex, hippocampus, cerebellum, basal glia, midbrain, thalamus, and glial cells [24,25,26,27,28]. CYP2D6 is also involved in the metabolism of tyramine to dopamine, in the regeneration of serotonin from 5-methoxytryptamine, and in behavioral traits and psychopathology [29,30,31,32,33].

### 2.2. Oxycodone Metabolism in Humans (PK and PGx; GRADE High Quality ++++)

Drug metabolism studies have characterized the extensive biotransformation of oxycodone in humans. Oxycodone is *N*-demethylated to noroxycodone and O-demethylated to oxymorphone; both undergo subsequent major metabolism by either *N*-demethylation (noroxycodone into noroxymorphone) or glucuronidation (oxymorphone into oxymorphone 3-glucuronide) [34,35]. In vitro studies conducted with human liver microsomes and genetically-engineered recombinant drug-metabolizing enzyme products have demonstrated that CYP3A4 and CYP2D6 isoforms mediate most of oxycodone metabolism [34,36]. A small amount is converted to α/β oxycodol via 6-keto reduction and a small fraction is also directly converted to a glucuronide conjugate (Figure 1) [37].

Lalovic et al., using human liver microsomes, showed that CYP3A4 was the highest-affinity enzyme involved in the *N*-demethylation of oxycodone to noroxycodone with a mean K_m_ of 600 ± 119 μM [34]; co-incubation with the CYP3A4 inhibitor ketoconazole decreased formation of noroxycodone by more than 90%. During the O-demethylation of oxycodone to oxymorphone, CYP2D6 was found to have the highest affinity, with a K_m_ of 130 ± 33 μM [34]. Other enzymes contributed to less than 8% for the *N*-demethylation and 10–26% for the O-demethylation pathways, respectively. *N*-Demethylation activity in intestinal microsomes was 20–50% that of the liver microsomes, while the O-demethylation was negligible, suggesting that the liver may be primarily responsible for first-pass metabolism, especially for the O-demethylation [34]. These results are in agreement with CYP3A4 and CYP2D6 protein levels measured in the human small intestine [20,38]. Romand et al. confirmed the important roles of CYP3A4 and CYP2D6 in the phase I transformation of oxycodone, while implicating UGT2B7 (K_m_: 762 ± 153 μM) and to a lesser extent UGT2B4 (K_m_: 2454 ± 497 μM) in the phase II metabolism of oxycodone [37].

Human mass-balance and pharmacokinetic studies confirmed that oxycodone is extensively metabolized in humans with only 10% of the drug excreted unchanged in urine [39,40,41]. In agreement with in vitro studies, oral administration of oxycodone showed that CYP3A4 is involved in the major metabolic clearance pathway with about 50% of the drug being *N*-demethylated to noroxycodone, while about 10% is O-demethylated by CYP2D6 to oxymorphone [41,42,43,44]. Following oral administration of oxycodone (15 mg single-dose immediate-release formulation) in 10 healthy individuals, (one CYP2D6 PM, one CYP2D6 UM, eight CYP2D6 activity NMs), average peak plasma concentrations of oxycodone reached approximately 30 ng/mL, while oxymorphone peak plasma concentrations were about 0.7 ng/mL [45,46,47,48]. In plasma of CYP2D6 NMs, the oxycodone-to-oxymorphone ratio could therefore be estimated at ~43:1 (Table 1) [46,47,48]. Under similar conditions, noroxycodone concentrations reached about 20 ng/mL [45,46,47,48].

Valuable information on the relative contributions of CYP2D6 and CYP3As to the disposition of oxycodone can also be obtained from pharmacogenomic and drug–drug interactions studies. Studies performed in individuals with multiple copies of CYP2D6—which translates to an UM phenotype—have shown that average plasma levels of oxycodone are reduced while average concentrations of oxymorphone are increased (relative to NMs). The oxycodone-to-oxymorphone plasma-concentration ratio was then estimated to be ~32:1 in UMs following oral administration (Table 1) [46,47,48].

In patients and healthy volunteers with non-functional CYP2D6 (PMs), the oxycodone-to-oxymorphone ratio is increased to approximately 300:1 (Table 1) [48,49]. This increased ratio is mostly explained by a significant decrease in oxymorphone plasma concentrations, as the overall contribution of CYP2D6 to the total clearance of oxycodone is limited. Still, a nominal amount of oxymorphone is detected in the plasma of these individuals, suggesting that an isoform other than CYP2D6 could also mediate the formation of oxymorphone. Similar results were obtained during in vitro drug metabolism studies [34]. Notably, much lower ratios of oxycodone/oxymorphone were observed in patients from all phenotypic groups following the administration of oxycodone intravenously [50]. This observation can be explained by a route of administration-dependent difference in the relative contributions of CYP3A4 and CYP2D6 to the clearance of oxycodone, as CYP2D6 expression is limited in the intestinal tissue [21,51,52,53]. Hence, the relative contribution of CYP2D6 to the total clearance of oxycodone appears increased following intravenous administration.

Drug interactions studies have been conducted with potent CYP2D6 inhibitors. Heiskanen et al. and Sirhan-Daneau et al. conducted studies in normal CYP2D6 metabolizers and demonstrated—using quinidine as a non-competitive and potent inhibitor of CYP2D6—that formation of oxymorphone was almost completely impeded under potent CYP2D6 inhibition [46,47,54,55]. However, using sensitive LC-MS/MS assays, limited amounts of oxymorphone could still be detected in their plasma [54]. Similar results were obtained by Lemberg et al. using paroxetine as a mechanism-based inhibitor of CYP2D6 [56]. Under potent inhibition of CYP2D6, the oxycodone-to-oxymorphone plasma-concentration ratio is nearly ~110:1 (Table 1).

As mentioned previously, CYP3A4 is the major enzyme involved in the disposition of oxycodone. Inhibition of CYP3A4, using inhibitory agents such as itraconazole or ketoconazole, is associated with increases in both oxycodone and also oxymorphone levels. As CYP3A4 is inhibited, more oxycodone is available for its metabolism by CYP2D6 into oxymorphone. Under conditions of CYP3A4 inhibition, the oxycodone to oxymorphone plasma concentration ratios ranged from 56:1 to 21:1 (Table 1) [57,58,59].

### 2.3. Oxycodone Distribution and Protein Binding (PK; GRADE Low Quality ++−−)

Average plasma protein binding for oxycodone is 45%, while that for oxymorphone is 11%. Hence, for all situations described above, the relative oxycodone/oxymorphone ratios while considering free-drug concentrations in the plasma [(C_avg oxycodone_ × F_u_)/(C_avg oxymorphone_ × F_u_)] should be decreased by about 40% (Table 1).

Drug distribution studies have demonstrated that both oxycodone and oxymorphone can cross the blood–brain barrier. First, Bostrom et al. reported in two different studies that the brain-to-blood unbound concentration ratio of oxycodone was 3.0–6.0 [60,61]. Then, Zasshi et al. also reported that the brain-to-blood unbound concentration ratio for oxycodone was 3.0 [62]. In regards to oxymorphone, Sadiq et al. reported that the brain-to-blood unbound concentration ratio for oxymorphone was 1.9 [63]. In contrast, Zasshi et al. and Lalovic et al. reported that the oxymorphone brain-to-blood unbound concentration ratio was 0.26 to 0.3 [34,44,62]. However, the discrepancy between these results could not be explained. Assuming a conservative brain-to-blood unbound concentration ratio of 3.0 for oxycodone and an average of 1.0 for oxymorphone, new oxycodone-to-oxymorphone ratios can be estimated (Table 1); noroxycodone and noroxymorphone do not have good brain penetration [44].

### 2.4. Oxymorphone Pharmacokinetics Following Direct Oxymorphone Administration (PK; GRADE Moderate Quality +++−)

Oxymorphone is a known potent μ-opioid receptor agonist which may contribute, totally or partially, to the overall analgesic effects observed following oxycodone oral administration [64]. Oxymorphone can be administered directly as an active drug to humans and its pharmacokinetics and pharmacodynamics can be appreciated and compared to its pharmacokinetics/pharmacodynamic relationship observed following oxycodone administration.

The bioavailability of oral oxymorphone is about 10% following an important first-pass metabolism. Oxymorphone undergoes extensive conjugation mediated mostly by UGT2B7 to produce its primary metabolite, oxymorphone 3-glucuronide [65,66]. To a lesser extent, it also undergoes reduction of the 6-ketone group to form 6-hydroxy-oxymorphone. After administration of a single 10 mg oxymorphone immediate-release oral dose, 1.9% was eliminated in urine as free oxymorphone, 44.1% as conjugated oxymorphone 3-glucuronide, 0.3% as 6-hydroxy-oxymorphone, and 2.6% as conjugated 6-hydroxy-oxymorphone [35]. Note that in contrast to other opioids, the *N*-demethylation of oxymorphone into noroxymorphone is very limited in humans [35].

Following the oral administration of a single extended-release 10 mg dose of oxymorphone, peak plasma concentrations of oxymorphone, 6-hydroxy-oxymorphone, and oxymorphone 3-glucuronide reached 0.65 ng/mL, 0.37 ng/mL, and 112 ng/mL, respectively [67]. Plasma levels of oxymorphone observed under these conditions were similar to plasma concentrations observed following administration of a single 15 mg immediate-release oxycodone dose (0.7 ng/mL) [46,47,55]. This observation strongly suggests that oxymorphone plasma levels of 0.5 to 1.0 ng/mL mediate clinically relevant analgesic effects and likely contribute significantly to oxycodone efficacy.

## 3. Pharmacodynamic Considerations

### 3.1. Interaction of Oxycodone and Its Metabolites with Opioid Receptors (PD; GRADE Moderate Quality +++−)

Oxycodone has the strongest affinity for the μ-opioid receptor (K_i_ = 18 nM) compared to the κ-(K_i_ = 677 nM) and δ-(K_i_ = 958 nM) opioid receptors, respectively [68,69]. Oxymorphone has been shown to have 10–60 times more affinity for μ-opioid receptors compared to oxycodone [55]. In vitro GTPγ binding studies using human μ-opioid receptors expressed in Chinese hamster ovary (CHO) cells have shown that oxymorphone may induce 30- to 40-fold greater increases in GTPγ binding than oxycodone [70]. In various rat brain regions, oxymorphone showed a 10- to 100-fold higher potency to activate GTPγ binding compared to oxycodone [71]; noroxycodone and noroxymorphone do not have significant μ-opioid agonist activity [72]. The same is true for oxymorphone 3-glucuronide, which is the major circulating metabolite following oxymorphone administration (almost a 200:1 ratio oxymorphone 3-glucuronide/oxymorphone) [67].

### 3.2. Unlikely Formation of Active 6-Glucuronide and Other Glucuronide Metabolites (GRADE Moderate Quality +++−)

As described by Wang et al., there is one hydroxyl group on oxycodone and two hydroxyl groups on the oxymorphone structure, making them potential targets for glucuronidation [73]. The hydroxyl group attached to carbon 14 is an alcoholic group, and this moiety is relatively stable and protected within the ring structure, creating steric hindrance (Figure 1). Therefore, glucuronidation is unlikely to occur for either oxycodone or oxymorphone [73]. This leaves only the hydroxyl group at position 3 of the oxymorphone structure to react with glucuronic acid—the major metabolite of oxymorphone—as a carbonyl instead of a hydroxyl group is present at position 6 of both oxycodone and oxymorphone (Figure 1). The formation of 6-hydroxy glucuronide metabolites from either oxycodone or oxymorphone is very limited as it requires prior reduction of the 6-keto moiety [35,37,44]. Contrary to codeine and morphine, which produce 6-glucuronide metabolites with known μ-opioid receptor agonist activities, oxycodone and oxymorphone 6-glucuronide metabolites are not expected to be formed to a significant extent and to contribute clinically to oxycodone activity.

### 3.3. Opioid μ-Receptor (OPMR1) and Catechol-O-Methyl Transferase (COMT) Genetic Polymorphisms and Response to Opioid Drugs (PD and PGx; GRADE Low Quality ++−−)

Studies generally use a battery of pain tests to gauge drug efficacy. Some commonly used tests include the cold pressor test (cardiovascular response and pain perception), the electrical stimulation test (detects pain from Aδ and C sensory nerves), and thermal (thermode/thermal grill) and mechanical tests. Patients often use the visual analog scale to mark their pain levels throughout the tests. When comparing studies, it is essential to compare the tests used to gauge pain thresholds, as types of testing could introduce potential confounding factors [74].

Oxycodone and oxymorphone exert their pharmacological effects mainly as agonists to μ-opioid receptors. These receptors are encoded by the *OPRM1* gene located on chromosome 6. Various polymorphisms in the *OPRM1* gene have been identified, which may be important in analgesic response, pathologies of addiction, and adverse events related to opioid use [75,76,77]. In this section, we review studies that included oxycodone alone or along with other opioids to assess the impact of *OPRM1* polymorphisms on drug actions.

The *OPRM1*
*A118G* (rs1799971) is one of the most widely studied variants in the context of opioid response. Gong et al. reported that higher opioid doses were required 24 h post-observation in cancer patients who were carriers of the *G* allele [78]. This polymorphism was also associated with progressively increasing 24-h opioid doses [78]. In an outpatient setting, a study by Lostch et al. described a tendency towards increased pain (decreased opioid response) with increasing number of *OPMR1*
*G* alleles [79]. Other studies report either a reduced analgesic effect of oxycodone in healthy subjects or the requirement of increased doses of oxycodone in patients carrying the *G* allele compared to individuals with an *OPRM1*
*AA* genotype [80,81]. Olesen et al. showed an association between oxycodone response and *OPRM1* polymorphisms using four different pain tests [74]. They found that each test was associated with different single nucleotide polymorphisms (SNPs) in the *OPRM1* gene. For example, the *G* variant was associated with oxycodone response in the visceral pain tolerance threshold. However, it was not associated with oxycodone response in either thermal skin pain tolerance, muscle pressure tolerance, or thermal visceral tolerance thresholds [74].

Two meta-analyses studied populations of 4607 and 5902 patients and found that carriers of the *OPMR1*
*G* allele—whether it was an *AG* or *GG* variant—required higher opioid doses and experienced higher pain scores [82,83]. In a prospective study in 153 opioid users, higher opioid consumption required for pain management was observed for carriers of the *G* allele [84]. Ren et al. also performed a meta-analysis and reported a reduction in nausea and vomiting in *G* allele carriers compared to *AA* carriers, with an odds ratio of 1.30 [83]. Other studies found no association between the *A118G* polymorphism and opioid-induced side effects [85]. In a study of 268 patients, it was reported that post-operative response to intravenous oxycodone was not associated with *OPRM1* G variant [86]. Other variants in the *OPRM1* gene have not been extensively linked to oxycodone response in the clinic.

The catechol-O-methyl transferase (COMT) is an enzyme involved in the metabolism of neurotransmitters and acts as a key modulator of dopaminergic and adrenergic/noradrenergic neurotransmission. The *COMT* gene is polymorphic and the most common c.472 *G > A* variant, in which the amino acid valine is substituted for a methionine, results in reduced COMT enzyme activity [87]. In human studies, the *COMT* genotype (Val/Met) has been shown, under different settings, to affect the efficacy of opioids in acute and chronic pain [88]. Low COMT activity has been associated with increased pain sensitivity, increased opioid analgesia, and increased opioid side effects, such as nausea and vomiting [85,87,89,90]. However, there is no consensus on the real clinical impact of this polymorphism on opioid response.

## 4. Pharmacokinetic, Pharmacogenomic, and Pharmacodynamic Considerations

As indicated by the CPIC’s position, it is debated whether or not *CYP2D6*, *OPMR1*, and other genetic polymorphisms contribute to the analgesic effects of oxycodone. Unfortunately, there are a limited number of well-designed studies that consider these covariables simultaneously. The CPIC guidelines recommend pharmacogenetic testing for structurally related opioid congeners, such as codeine and tramadol, which are also prodrugs activated by CYP2D6 into their respective active metabolites (morphine and O-desmethyltramadol, both of which have free hydroxyl groups at position 3). These metabolites are also more potent μ-opioid receptor agonists than their respective parent prodrugs [4]. In the next sections, we report information from pharmacokinetic, pharmacodynamic, and pharmacogenetic studies that mostly lack pharmacogenomic information for the *OPMR1* or *COMT* polymorphisms and their impact on the analgesic efficacy of either oxycodone or oxymorphone.

### 4.1. Pharmacogenetic Studies Relating Oxymorphone Formation to Pain Management by Oxycodone (PK, PD, and PGx; GRADE High Quality +)

Zwisler et al. conducted a double-blinded, randomized, placebo-controlled study of healthy subjects to assess oxycodone analgesic effects in PMs and NMs of CYP2D6 [91]. They demonstrated that PMs had an attenuated response to oxycodone compared to NMs when using both the cold pressor test and the electrical stimulation tests [91]. As expected, PMs also had a higher oxycodone/oxymorphone ratio, suggesting that CYP2D6-mediated formation of oxymorphone is important for anti-nociceptive effects [91]. Samer et al. tested the efficacy of oxycodone in patients with different CYP2D6 genotypes [64]. They used a series of pain tests to show that, after administration of oxycodone, UMs had a 1.4-fold higher tolerance threshold to the cold pressor test, whereas PMs had a 20-fold lower threshold when they were compared to NMs [48]. Similar results were seen with the electrical stimulation test, where respiratory depression and sedation caused by oxycodone were also found to be significantly higher in UMs compared to NMs. Unambiguously, all UMs exhibited mild to severe side effects whereas PMs did not report any [48]. In that study, the pain thresholds were significantly associated with oxymorphone and noroxymorphone concentrations, whereas there was no association between oxycodone and noroxycodone concentrations and effects observed [64]. These authors concluded that oxycodone does not appear to be the active molecule responsible for the analgesic effects [48].

Results from these studies clearly point towards an important role for CYP2D6-mediated oxycodone transformation into oxymorphone in pain control. Other evidence also comes from case reports [92,93,94]. For instance, a patient with a fractured hip and a history of codeine intolerance was prescribed a regimen of oxycodone and acetaminophen [92]. As a proper level of analgesia was not achieved, she then received several doses of tramadol, as needed. Even with the added tramadol, the patient obtained no pain relief. Later, her genotype results confirmed that she was a CYP2D6 PM [92]. This suggests that, as for codeine and tramadol, oxycodone depends on CYP2D6 for its analgesic effects. Tyndale et al. also reported that CYP2D6 PMs had a lower probability of developing dependence on oxycodone compared to NMs; however, this was mostly explained by limited exposure to opioid agonists in this subset of the population [95].

### 4.2. Drug–Drug Interaction Studies Modulating CYP2D6 Activity to Assess Oxymorphone Contribution to Oxycodone Efficacy (PK, PD, and PGx Plus DDIs; GRADE Moderate Quality +++−)

Drug–drug interaction studies are another way to obtain valuable insights into the contribution of oxymorphone to oxycodone actions. CYP2D6 has a limited role (10%) in the total systemic clearance of oxycodone, and modulation of CYP2D6 activity is not expected to be associated with significant changes in oxycodone plasma concentrations. However, CYP2D6 is largely responsible for converting oxycodone to oxymorphone. Hence, drug–drug interaction studies using selective and potent CYP2D6 inhibitors could help assess the oxymorphone contribution to oxycodone pain control actions. Both quinidine (which does not seem to cross the blood–brain barrier) and paroxetine (which can enter the brain) have been used as CYP2D6 inhibitors in drug interaction studies.

As mentioned previously, Samer et al. conducted a study in 10 healthy subjects to assess the impact of quinidine pretreatment on oral oxycodone and assess multiple pharmacodynamic parameters, where phenotyping for CYP2D6 activity was conducted using dextromethorphan as a probe drug [64]. Quinidine pretreatment blunted oxycodone-related pharmacodynamic effects on pain thresholds, such as the nociceptive flexion reflex (objective NFR), the subjective pain threshold (SPT) following an electrical stimulus, and sedation, making these effects similar to measurements observed under placebo pretreatment (no oxycodone). However, quinidine pretreatment did not completely abolish all pharmacodynamic effects, particularly in the pupil size and cold pressor test assessments. These effects may be related to the local metabolism in the brain of oxycodone into oxymorphone and the fact that quinidine does not readily penetrate the blood–brain barrier and might not be efficient at blocking CYP2D6 in the CNS (compared to its high potency in the liver) [55,64]. Pharmacodynamic results obtained by Samer et al. support the notion that oxymorphone production is required for oxycodone efficacy [64].

Heiskanen et al. tested the pharmacokinetics and pharmacodynamics of oxycodone administered with either placebo or quinidine pretreatments in ten healthy volunteers that were determined to be NMs using a phenotyping test (debrisoquine as a CYP2D6 probe drug) [55]. There was only a small reduction of the oxycodone AUC_0–24_ following quinidine pretreatment compared to a placebo (258.5 ng.h/mL vs. 228.8 ng.h/mL). However, oxymorphone exposure decreased by 98% when quinidine was given three hours before oxycodone and six hours after to account for extended release of oxycodone, pointing to the importance of CYP2D6 activity in oxymorphone production [55]. Several drug effects were measured through questionnaires, accompanied by pupil size measurements and psychomotor tests (like the Maddox wing test). In the placebo pretreatment arm, several of the drug effects observed including alertness, pain control, and sleepiness, correlated with both oxycodone and oxymorphone plasma concentrations [55]. Prevention of the production of oxymorphone by quinidine (oxymorphone concentrations were undetectable in 8/10 patients) did not affect the psychomotor or subjective drug effects of oxycodone, suggesting that oxycodone, not oxymorphone, was responsible for drug effects. However, the authors noted that the interpretation of their results could lead to different conclusions if a potential metabolism of oxycodone into oxymorphone in the brain was to be considered [55].

Using another CYP2D6 inhibitor, paroxetine, Kummer et al. studied the effects of another CYP2D6 inhibitor—paroxetine, on oral oxycodone—in 12 healthy participants [96]. They found that the oxycodone C_max_ and AUC_0-∞_ were unchanged after paroxetine pretreatment, as expected, since CYP2D6 does not contribute significantly to oxycodone metabolism [96]. However, they reported that paroxetine prevented the analgesic activity of oxycodone when assessed using the cold pressor test, whereas pretreatment with ketoconazole (CYP3A4 inhibition, to be discussed later) accentuated oxycodone’s analgesic effects (area under the effect curve and the values for the maximal effect Emax, *p* < 0.05 vs. paroxetine) [96]. Similar trends were observed on pupillometry effects. Their observation supports the notion that oxymorphone plays an important role in the analgesic effects following oxycodone administration. However, the authors suggested that, as a limitation to their study, selective serotonin inhibitor drugs such as paroxetine may interfere with the interpretation of pharmacodynamic measurements such as the cold pressure test and pupillary reaction [96].

Finally, Lemberg et al. used paroxetine coadministration in a cohort of 20 chronic pain patients to determine effects of CYP2D6 blockade on oxycodone pharmacokinetics and drug effects. Notably, this study included patients who were both naïve and non-naïve to opioids [56]. Patients were included in the study if they did not need more than two rescue doses of morphine in a day for at least three days. Patients were genotyped for CYP2D6, resulting in 14 NMs, 4 IMs, and 2 UMs; no PMs were reported. Paroxetine coadministration significantly increased the C_max_ (26%) and AUC_0–12_ (19%) of oxycodone, while significantly decreasing the C_max_ (−57%) and AUC_0–12_ (−67%) of oxymorphone [56]. Though statistically significant, changes in oxycodone pharmacokinetics were less prominent than the changes in oxymorphone pharmacokinetics, as expected. Drug effect was measured as the quantitative increase in morphine dose required to manage breakthrough pain on oxycodone treatment. In this study, no differences in the additional morphine doses required as rescue medication or in the visual analog scale of pain were observed between paroxetine pretreatment vs. a placebo [56]. Hence, it was concluded that paroxetine did not affect oxycodone-associated analgesia. Note that variability in the morphine dose needed at baseline and the use of standard morphine rescue doses could potentially mask reduction in anti-nociceptive activity during paroxetine cotreatment [56].

Additional studies using quinidine or paroxetine as potent CYP2D6 inhibitor showed that oxymorphone levels were significantly reduced by the coadministration of these drugs, but discrepancies existed regarding changes in pharmacodynamic parameters when oxycodone was compared to oxycodone plus CYP2D6 inhibitors [49,97]. Assessment of pain levels and response to medication remains a semi-quantitative and somewhat subjective measure that could be influenced, either positively or negatively, by several external experimental and individual factors. Nevertheless, results from these studies must be considered in the overall determination of the relative contribution of oxymorphone to oxycodone effects in pain management [49,97]. Figure 2 provides a summary of the relationship between oxycodone and oxymorphone pharmacokinetics, *CYP2D6* genetic polymorphisms, and pharmacodynamics.

### 4.3. Drug–Drug Interactions Studies Modulating CYP3A4 Activity to Assess Oxymorphone Contribution to Oxycodone Efficacy (PK-DDI and PD; GRADE Low Quality ++−−)

Another way to try to dissect oxymorphone from oxycodone actions is to modulate CYP3A4 activity by administrating inhibitors such as ketoconazole, itraconazole, or voriconazole or inducers like rifampin [45]. Using electrical stimulation and cold pressor tests, Samer et al. demonstrated that ketoconazole coadministration increased all pharmacodynamic effects of oxycodone, including sedation and the pain threshold [64]. They also performed a pharmacokinetic–pharmacodynamic multivariate analysis to identify pharmacokinetics predictors (including oxycodone, oxymorphone, noroxycodone, and noroxymorphone) associated with outcomes (area under the drug effect time curve: AUEC_90_). The only positive predictor of the subjective pain threshold and of oxygen saturation was oxymorphone. CYP3A4 blockade with ketoconazole was associated with an increased risk of adverse effects, most notably in CYP2D6 UMs [64].

Using itraconazole to inhibit CYP3A4, Saari et al. demonstrated that inhibition of CYP3A4 significantly increased the concentration of intravenous oxycodone by 51% and of oral oxycodone by 125% [58]. This finding was expected since CYP3A4 is expressed both in the intestine and in the liver such that itraconazole inhibition can occur in both tissues following oral administration of oxycodone. It was also shown that oxymorphone plasma concentrations increased even further (159% (intravenous) and 359% (oral), respectively) [58]. Itraconazole coadministration had a significant effect on behavior (alertness, deterioration of performance, etc.) following the oral administration of oxycodone. In contrast, no significant differences in the subjective drug effect or drowsiness were observed following the coadministration of itraconazole with intravenous oxycodone [58]. A relationship between oxymorphone plasma concentrations and pain control was not observed in their study. Interestingly, oxycodone, when administered alone, did not significantly increase the heat-pain threshold in all subjects; itraconazole coadministration also did not modify this outcome. This finding underlines that pain assessment, being subjective, is challenging in healthy volunteers since pain includes psychological, behavioral, and neurological aspects [58].

Voriconazole was also used to assess effects of CYP3A4 modulation on oxycodone-mediated analgesia in healthy volunteers [57]. There was a significant increase in concentrations for both oxycodone and oxymorphone (257% and 597%, respectively) [57]. Voriconazole coadministration increased some pharmacodynamic effects, including heterophoria and myosis, when compared to placebo administration. However, voriconazole did not alter the heat-pain and cold-pain thresholds or the cold-pain intensity of oxycodone. Voriconazole coadministration did not significantly alter the perceived subjective drug effect (VAS) or other observed behaviors [57]. Similar to the study by Saari et al. using itraconazole [58], effects of the variable *CYP2D6* genotype were not considered while interpreting the results of CYP3A4 inhibition using voriconazole (due to a small sample size) [57]. In the studies by Kummer et al. and Gronlund et al., both conducted among healthy volunteers, behavioral effects were recorded using questionnaires, and only the cold pressor test was conducted as a pain test [96,98]. Watanabe et al. reported a case series of patients with cancer in whom oxycodone-induced adverse events including drowsiness, vomiting, and hypopnea were reported within a few days after voriconazole treatment initiation [99]. The observed side effects were related to CYP3A4-mediated inhibition of oxycodone metabolism by voriconazole; however, the contribution of oxymorphone to the observed symptoms could not be excluded [99].

Gronlund et al. investigated the effects of the inhibition of CYP2D6 by paroxetine and the dual inhibition of CYP2D6 and CYP3A4 by paroxetine and itraconazole on the pharmacokinetics of and pharmacological response to intravenous oxycodone [100]. The administration of paroxetine did not significantly alter oxycodone exposure, but oxymorphone exposure was decreased by 60% [100]. Surprisingly, even though the combined administration of paroxetine and itraconazole increased oxycodone exposure by twofold, it did not cause any changes in pharmacological response. However, this study had some limitations, as *CYP2D6* genotypes were not considered, a phase with CYP3A4 inhibition alone was not conducted (only a cold-pain test was used), and the power of the study design was not sufficient for detailed pharmacodynamic analyses [100].

Together these studies clearly indicate that blockade of CYP3A4 is associated with significant increases in oxycodone and oxymorphone concentrations. However, the respective contributions of oxycodone and oxymorphone to the pharmacological response, when observed, could not be distinguished. When CYP2D6 and CYP3A4 activities were combined, effects on analgesia and side effects were observed if *CYP2D6* genetic polymorphisms were considered.

## 5. Other Covariables

### 5.1. Phenoconversion as a Confounding Factor (GRADE Low Quality ++−−)

In a real-world scenario, genotype-guided phenotypes can be altered by various environmental conditions such as concomitant medications and disease states, a phenomenon called “phenoconversion” [101]. Consideration of phenoconversion is essential when trying to estimate the effects of *CYP2D6* genotypes on drug efficacy. For instance, an individual with a *CYP2D6 *1/*1* genotype predicted to be a NM could rather behave phenotypically as a PM of CYP2D6 for some substrates if this individual is administered concomitant treatment with other CYP2D6 substrates or inhibitors [102]. Similarly, a patient with type 2 diabetes may show diminished CYP3A4 activities and decreased clearance of CYP3A substrates due to downregulation of CYP3A4 expression secondary to chronic inflammatory conditions [101,103]. Phenotypic determination by the administration of probe drug substrates or by the use of endogenous biomarkers could allow for a more precise determination of functional enzyme levels than the genotype-predicted phenotype [104,105,106].

Zwisler et al. conducted a study in which oxycodone was administered intravenously to 270 patients for 24 h post-operatively, while morphine was prescribed as rescue medication [97]. The authors reported no association between oxycodone consumption and *CYP2D6* genotypes, though surgery may represent a potential trigger for systemic inflammation that could induce a phenoconversion. Notably, no phenotyping was derived and results were based only on genotype data, introducing a potentially confounding factor in the data interpretation. In addition, all patients were also treated with acetaminophen (paracetamol; 1000 mg four times a day) and diclofenac (50 mg three times daily), which may have blunted the overall appreciation of analgesic response associated with oxycodone. Patients taking a CYP2D6 inhibitor were excluded from the study; however, CYP2D6-inhibiting drugs were limited to paroxetine, fluoxetine, and terbinafine [97]. We recently published a study in which we found that 15% of patients taking CYP2D6-metabolized opioids were exposed to significant CYP2D6 drug–drug interactions [107]. It can be expected that patients with *CYP2D6*-induced phenoconversion may be underestimated. Furthermore, the *CYP2D6* alleles panel used in the study by Zwisler et al. was limited to four alleles, and neither *CYP2D6* gene deletion (**5*) nor gene duplication were tested [97]. Obviously, based on new, current knowledge, some patients were likely assigned to an inappropriate phenotype, as the use of a limited PGx panel can lead to a misclassification of the predicted phenotype based on the genotype.

The same research group assessed the analgesic effects of oxycodone in 33 healthy subjects using tramadol as a CYP2D6 probe substrate [91]. In addition to the phenotypic determination based on tramadol disposition, individuals were also genotyped for *CYP2D6* using the same panel test with limited *CYP2D6* allele determination [91]. They found, in five individuals, a mismatch between the determined genotype and observed phenotype based on tramadol pharmacokinetics. These five subjects were re-phenotyped using the gold-standard probe-drug sparteine. Phenotyping results with sparteine confirmed their PM phenotype as in agreement with the observed tramadol phenotype test but in disagreement with the determined genotype [91]. Individuals were healthy and were not allowed to take medications (except oral contraceptives), thus excluding disease-induced and drug-induced phenoconversion [91]. The discrepancy between the genotype-predicted phenotype and the observed phenotype can be explained by the limited pharmacogenetic panels tested, which created mis-assigned subjects in studied groups.

In the study conducted by Samer et al., individuals’ phenotype determinations using dextromethorphan as a probe for CYP2D6 were undertaken to assign metabolizer status [64]. In this study, 32 *CYP2D6* alleles were tested and subjects were healthy and were not taking concomitant medications. There was no mismatch between the predicted phenotype based on the *CYP2D6* genotype and the observed phenotype using the metabolic ratio of dextromethorphan. This study showed an association between CYP2D6 activity and oxycodone efficacy [64].

Drug-induced phenoconversion can blunt the effects of pharmacogenetics. The impact of concomitant medications on phenotype interpretation was not considered in the analyses of some studies [93,108]. For instance, in the study by Andreassen et al., a proportion of NMs were taking CYP2D6 inhibitors, which likely resulted in their phenoconversion to PMs [49]. In the analysis of their results, the authors grouped everyone based on their genotype, not their phenotype, potentially skewing the results.

### 5.2. Liver vs. Brain CYP2D6 Activity (GRADE Low Quality ++−−)

CYP2D6 is expressed and functional in neurons of various regions of the brain [109]. The concept of local formation of oxymorphone from oxycodone, independent of liver CYP2D6 activity, could in part explain the apparent lack of association between oxymorphone plasma levels and observed clinical response following oxycodone administration. When conducting a drug–drug interaction study using quinidine as a potent CYP2D6 inhibitor, we demonstrated phenoconversion of CYP2D6 NMs into PMs based on the pattern of oxycodone, noroxycodone, and oxymorphone plasma levels [46,47]. Partial metabolic clearances clearly demonstrated significant inhibition of CYP2D6 and oxymorphone plasma concentration values similar to those observed in genetically determined CYP2D6 PMs. However, we could not demonstrate a significant increase in response to pain stimuli (thermal cathode, cold pressure) following quinidine administration [46,47].

Similar results were observed by Heiskanen et al. [55,64]. Inhibition of P-glycoprotein by a potent inhibitor, such a tariquidar, demonstrated that the distribution of quinidine to the brain is limited by this efflux transporter (although quinidine itself could competitively inhibit P-glycoprotein) [110]. Consequently, while low doses of quinidine are able to inhibit liver CYP2D6, brain concentrations of quinidine achieved under these conditions may leave brain CYP2D6 functional and allow for local formation of oxymorphone; plasma concentrations of oxymorphone may not reflect its extracellular concentrations in the brain.

Paroxetine has also been shown to inhibit oxycodone-mediated CYP2D6 metabolism and formation of oxymorphone [56,98]. As paroxetine distributes extensively to the brain, observation of the inhibition of CYP2D6 in the liver, in addition to the brain, would be expected [111]. Gronlund et al. investigated in 11 healthy subjects the effects of CYP2D6 inhibition by paroxetine on the pharmacokinetics and pharmacological response of oral oxycodone [98]. Paroxetine had no significant effect on the pharmacological response to oxycodone compared to the placebo phase. It should be noted that *CYP2D6* genotypes were not considered even though PM and IM *CYP2D6* genotypes were found in three subjects [98]. The co-administration of paroxetine is not expected to have any significant impact in CYP2D6 PM or, to a lesser extent, in IM. Considering this limitation, this study likely did not have the ability to evaluate the effects of paroxetine on oxycodone. In addition, the effects of paroxetine itself on pain perception could be a confounding factor, as suggested previously [96,98,112].

Conceptually, PMs of CYP2D6 can express CYP2D6 in neither their livers nor in their brains. Strong association between oxymorphone levels and oxycodone effects in subjects with various *CYP2D6* genotypes supports the notion that oxycodone actions are largely mediated by oxymorphone. As other CYP isoforms appear to be able to form a certain amount of oxymorphone from oxycodone, this may also explain some discrepancies between study results.

## 6. Conclusions

Our systematic analysis of the current knowledge pertaining to the contribution of oxymorphone to oxycodone efficacy for pain management and the relevance of *CYP2D6* and *OPMR1* genetic polymorphisms lead us to the following deductions:(1)In vitro characterization of the CYP450 isoenzymes involved in the transformation of oxycodone into oxymorphone clearly demonstrated the predominant, but not exclusive, role of CYP2D6 [34,36]. Grade: high quality ++++;(2)Pharmacogenomic and drug–drug interaction studies conducted in humans also confirmed the role of CYP2D6 in the formation of oxymorphone. Grade: high quality ++++;(3)Plasma concentrations of oxymorphone observed following oral administration of oxycodone, although in the low nanogram level, are of the same magnitude as those measured when oxymorphone is directly administered, and they have been shown to be efficacious for pain management in humans. This suggests that oxymorphone mediates most of the analgesic efficacy of oxycodone. Grade: high quality ++++;(4)Binding studies at the μ-opioid receptor level have clearly established the much greater agonist affinity of oxymorphone vs. oxycodone [44]. Grade: high quality ++++;(5)The estimated brain concentration ratio of oxycodone vs. oxymorphone, considering protein binding and brain distribution (tissue level) and weighted for the relative potency of both compounds as agonists on the μ-opioid receptor, suggests that oxycodone and oxymorphone contribute equally to analgesic effects. Grade: low quality ++−−;(6)As the μ-opioid receptor is on the extracellular side of the neuron membrane in the synapse, the extracellular concentrations of oxycodone and oxymorphone could be more relevant than brain tissue levels. Unfortunately, these data are not available at the moment. Grade: very low quality +−−−;(7)Brain CYP2D6 activity, capable of converting oxycodone into oxymorphone, could explain the lack of association between oxymorphone plasma levels and efficacy observed in some drug–drug interactions and clinical studies. Grade: low quality ++−−;(8)The lack of oxycodone activity in genetically characterized CYP2D6 PMs, who functionally lack CYP2D6 activity not only in their liver but also in their brain, strongly supports the concept that oxycodone activity is mediated by oxymorphone. Grade: high quality ++++;(9)Clinically, the coadministration of other CYP2D6 substrates and inhibitors, and their capacity to blunt oxycodone efficacy, could be related to both their capacity to inhibit liver CYP2D6 activity and also to their capacity to cross the blood–brain barrier, penetrate CNS, and reach neurons and inhibit brain CYP2D6 activity, thus limiting local oxymorphone formation from oxycodone. Grade: low quality ++−−;(10)Finally, mutations on *OPMR1* leading to less functional μ-opioid receptors and their role in the inter-individual response to active opioid products is too often overlooked. Grade: low quality ++−−.

We support the current CPIC position as it offers stringent recommendations to establish guidelines regarding pharmacogenetic testing for oxycodone treatment. Nevertheless, clinical considerations and scientific information lead us to recommend that a phenotypic assessment of CYP2D6 activity should be considered when oxycodone is being prescribed for pain management. This position is supported, at least in part, as PGx testing is recommended for other opioid congeners (codeine, tramadol) that share several similar pharmacokinetic, pharmacodynamic, and pharmacogenomic properties. Phenotypic assessment should include the use of advanced clinical decision support systems to conduct comprehensive medication reviews and identify potential multi-drug interactions leading to CYP2D6 phenoconversion. Pharmacogenomic testing should also be considered, as it could provide a definitive response for CYP2D6 PMs, i.e., about 10% of patients with polypharmacy. Furthermore, more than 100 variant alleles have been described for *CYP2D6*, and the precision of the genotype results is directly related to the quality of the genotype test being performed. As directly active opioids are a clinically sound alternative to CYP2D6-activated opioids (oxycodone, codeine, hydrocodone, and tramadol), use of these agents instead of the prodrugs appears highly justifiable in patients with polypharmacy or with suspected non-functional or decreased CYP2D6 activity.

A summary of the studies included in this review study is presented in Table 2.

## Figures and Tables

**Figure 1 pharmaceutics-13-01466-f001:**
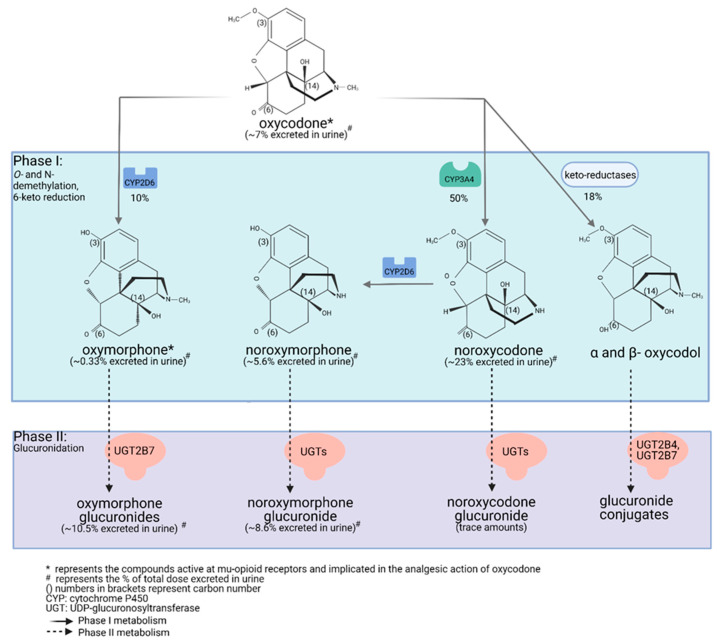
Detailed illustration of the elimination pathways of oxycodone. Created with Biorender.com (access date: 10 May 2021).

**Figure 2 pharmaceutics-13-01466-f002:**
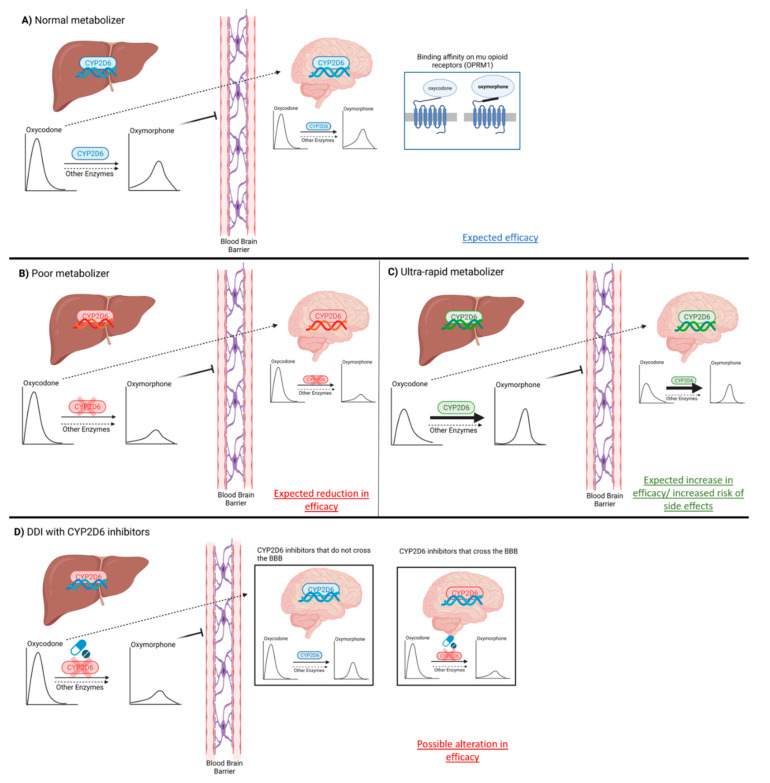
Implications of pharmacokinetics (including *CYP2D6* genetic polymorphism and CYP2D6 drug–drug interactions) and pharmacodynamics as determinants of interindividual difference in drug response to oxycodone and oxymorphone. CYP2D6: cytochrome P450 2D6; DDI: drug–drug interaction; BBB: blood–brain barrier. Created with Biorender.com (access date: 10 May 2021).

**Table 1 pharmaceutics-13-01466-t001:** Change in oxycodone:oxymorphone ratio depending on patient genotype or use of CYP450-inhibitor drugs.

CYP2D6 Activity	Oxycodone/Oxymorphone Concentration Ratio in Plasma	Oxycodone/Oxymorphone Free-Drug Concentration Ratio in Plasma	Oxycodone/Oxymorphone Free-Drug Concentration Ratio in the Brain *	Oxycodone/Oxymorphone Relative Contribution to μ-Opioid Receptor-Binding Considering the Free-Drug Concentration Ratio in the Brain **
UM CYP2D6	32:1	19:1	57:1	0.6:1
NM CYP2D6	43:1	26:1	78:1	0.8:1
PM CYP2D6	300:1	180:1	540:1	5.4:1
With potent CYP2D6 inhibitors	110:1	66:1	198:1	2:1
With potent CYP3A4 inhibitors	56:1 to 21:1	34:1 to 13:1	102:1 to 39:1	1:1 to 0.4:1

CYP2D6: cytochrome P450 2D6; UM: ultra-rapid metabolizer; NM: normal metabolizer; PM: poor metabolizer; CYP3A4: cytochrome P450 3A4; *: assuming a conservative brain-to-blood unbound concentration ratio of 3.0 for oxycodone and of 1.0 for oxymorphone; **: assuming a conservative brain-to-blood unbound concentration ratio of 3.0 for oxycodone and of 1.0 for oxymorphone and a potency ratio of 1:100 for oxycodone vs. oxymorphone.

**Table 2 pharmaceutics-13-01466-t002:** Summary of studies reviewed in this text with their design, measures, and outcomes.

Studies	Study Design	PK, PD, and/or PGx Measures	Outcomes	Conclusion
*CYP2D6* polymorphisms: Effect on oxycodone pharmacokinetics and pharmacodynamics				
Only PGx measurements				
Zwisler et al. (2009) [91]	33 healthy volunteers were administered 10 mg oral OXY or a placebo in a crossover design	PK measures: OXY, OXM, plasma concentrations (C_max_), and OXM/OXY ratio were recorded PD measures: Nociception was tested using electrical sural nerve stimulation and cold pressor tests; side effects were self-reported and rated on a verbal scale PGx measures: CYP2D6 genotype was determined using 4 alleles; all volunteers underwent additional phenotyping using tramadol as a probe to determine metabolizer status	OXM/OXY plasma ratio reduced by 58% in PMs compared to NMs (*p* < 0.001) OXY significantly induced analgesia in both NM and PM PMs were less sensitive to nociceptive tests than NMs; both groups reported side effects	OXM may be important for anti-nociception mediated by OXY
Zwisler et al. (2010) [97]	270 patients undergoing thyroid surgery or hysterectomy received IV OXY and morphine (rescue medication). The study was open label; however, CYP2D6 genotypes were double-blinded	PK measures: Plasma concentration of OMY and OXM were measured 30 min after OXY bolus dose PD measures: Pain was rated using numerical rating scale by patients; opioid side effects and sedation level were rated by study staff PGx measures: Four alleles were used to genotype patients for *CYP2D6*; no phenotyping was conducted	When PM were compared with NM, OXY levels were unchanged and OXM plasma concentration was significantly lower OXM/OXY ratio was reduced by 73% in PMs (*p* < 0.0001) OXY consumption, response to OXY, pain ratings, and side effects did not differ between genotypes	OXM may not be important for anti-nociception based on genotype grouping Caveats: Phenotyping was not conducted, **5* allele was not analyzed
Samer et al. (2010) [64]	10 healthy volunteers were administered oral OXY (0.2 mg/kg) with or without the CYP3A4 inhibitor ketoconazole and CYP2D6 inhibitor quinidine in a randomized crossover design	PD measures: Nociceptive tests including flexion reflex, the cold pressor test, thermal perception, light reflex of pupil, and psychomotor tests. Sedation was recorded as a side effect PGx measures: Genotyping was conducted for CYP2D6 (32 alleles) and phenotyping for both CYP2D6 and CYP3A4 (using dextromethorphan and midazolam, respectively) was conducted	All nociceptive tests strongly correlated with CYP2D6 activity, but not CYP3A4 activity; PMs had lower anti-nociception and higher sedation after OXY compared to EMs and UMs DDI studies showed a significant PK–PD correlation between OXM and norOXM; no correlation was observed with OXY and norOXY and the nociceptive test. Pupil size tests also showed a correlation with OXY	
Andreassen et al. (2012) [49]	2294 patients with malignant pain from multiple centers who had *CYP2D6* PGx data available were selected for analyses; OXY was administered by SC, orally, or IV	PK measures: OXY, norOXY, OXM, and norOXM concentrations were measured in blood as a one-time measurement PD measures: Pain, cognition, functional status, quality of life, and adverse events were measured PGx measures: Five alleles were tested for *CYP2D6* genotype, including gene duplication	Steady state concentrations of OXY and norOXY were not different between groups; however, OXM and norOXM concentrations were significantly higher in UMs and EMs compared to PMs There were no differences between any groups in the pharmacodynamic measures	OXM may not be important for analgesia Caveats: No phenotyping, limited sample sizes, no phenoconversion consideration for concomitant CYP2D6 and 3A4 medicine
**No PGx; only DDI measurement**				
Kummer et al. (2011) [96]	14 healthy volunteers, all NMs based on genotyping, were administered 0.2 mg/kg oral OXY with or without ketoconazole and paroxetine	PK measures: OXY, norOXY, and OXM were measured in blood PD measures: Pupil diameter, cold pressor test measurements, and adverse events using a visual analog scale	Ketoconazole significantly increased the AUC_0–∞_ of oral oxycodone, whereas paroxetine did not affect PK; ketoconazole also decreased the C_max_ of noroxycodone, and paroxetine decreased the C_max_ of oxymorphone significantly Pupil dilation was accentuated by ketoconazole and inhibited by paroxetine pretreatment Ketoconazole pretreatment accentuated OXY-mediated analgesia, whereas paroxetine inhibited this effect	OXM formation may be essential for OXY mediated analgesia
**PGx and DDI measurements**				
Heiskanen et al. (1998) [55]	10 healthy volunteers were administered 20 mg oral OXY either with a placebo or quinidine	PK measures: Plasma concentrations of OXY, norOXY, and OXM were measured PD measures: Drug effects and side effects were measured on a visual analog scale, psychomotor tests—e.g., the Maddox wing test, digital symbol substitution test, critical flicker fusion test, and pupillometry—were also conducted PGx measures: CYP2D6 phenotypes were assessed using debrisoquin as a probe drug; PMs were excluded	OXM formation was blocked by quinidine pretreatment; norOXY formation was significantly increased, but there was no significant difference in OXY exposure PD measures including drug effect and pupil size correlated with OXY and OXM plasma concentrations. However, no difference was observed in PD effects between the placebo and quinidine pretreatments Side effects were not different between quinidine and placebo pretreatments	OXM may not be responsible for psychomotor effects Caveats: No pain tests were carried out to determine analgesic effects
Hagelberg et al. (2009) [57]	12 healthy volunteers were given oral OXY and co-administered either a placebo or voriconazole (CYP3A4 inhibitor)	PK measures: OXY, norOXY, OXM, and norOXM concentrations were measured PD measures: Psychomotor tests, pupil diameter, and analgesic tests were conducted PGX measures: *CYP2D6* genotypes were determined; 11 alleles and gene duplication were determined	Voriconazole coadministration significantly reduced plasma exposure of OXY, norOXY, and norOXM, but increased exposure of OXM Voriconazole pretreatment did not significantly alter perceived drug effects, analgesia, or total reported adverse events	OXM does not seem to mediate analgesia Caveats: There was no sub-analysis of *CYP2D6* PGx, since PM would not have been affected by CYP3A4 inhibition
Lemberg et al. (2010) [56]	20 patients with pain were administered oral OXY and instructed to take morphine for breakthrough pain. After a stable dose was achieved, patients were instructed to take either paroxetine or a placebo. The change in the morphine dose requirement for breakthrough pain was measured	PK measures: OXY, norOXY, OXM, and norOXM were measured in blood PD measures: Pain intensity and pain relief using visual analog and verbal rating scales, drug effect using the modified drug effect scale, and adverse effects using a questionnaire were reported. Use of rescue medication was reported PGx measures: *CYP2D6* genotypes were assessed using six alleles	Paroxetine pretreatment significantly increased OXY (19%) and norOXY (100%) exposure and decreased OXM (−67%) and norOXM (−68%) exposure No differences were found in adverse events or analgesia between paroxetine and placebo phases, and no correlation was established between PK and genotype	OXM may not be important for pain control Caveats: No phenotyping; no poor metabolizers were found in the patient population (the group that shows the highest inefficacy)
Saari et al. (2010) [58]	10 healthy volunteers were given a placebo or itraconazole followed by IV or oral OXY in a four-way crossover design	PK measures: OXY, norOXY, OXM, and norOXM concentrations were measured PD measures: Behavioral tests, pupil diameter, and analgesic tests were conducted PGx measures: *CYP2D6* genotypes were determined; 11 alleles and gene duplication were assessed	Itraconazole reduced exposure of IV OXY and norOXM, while increasing exposure of norOXY Conversely, itraconazole increased exposure of oral OXY and reduced norOXY and norOXM Itraconazole pretreatment increased the perceived drug effect, reduced performance, and decreased pupil size Itraconazole pretreatment did not significantly alter analgesic effect measured by the heat-pain threshold; however, it significantly improved the cold-pain threshold Subjective drug effect correlated with plasma OXY and OXM concentration	OXY and OXM may both be important for drug effect; OXY-mediated analgesic effect depends on the test used to measure it Caveats: PGx sub-analysis was not conducted due to the small sample size in each group; there were no PMs in the study
Gronlund et al. (2010) [98]	11 healthy subjects were administered a placebo or oral OXY, alone or in combination with paroxetine (CYP2D6 inhibitor) or with paroxetine plus itraconazole (CYP2D6 and CYP3A4 inhibitors)	PK measures: OXY, norOXY, OXM, and norOXM concentrations were measured PD measures: Pain assessment, pupil diameter, and analgesic tests were conducted PGx measures: *CYP2D6* genotypes were determined; 11 alleles and gene duplication were determined	OXY exposure was not altered with paroxetine pretreatment alone, whereas OXM and norOXM exposure was significantly altered; itraconazole plus paroxetine pretreatment, however, increased plasma exposure by 2.77-fold on average Paroxetine pretreatment alone attenuated the effects of OXY on visual analog scales for performance deterioration Itraconazole plus paroxetine attenuated all PD variables for OXY; however, these differences were not statistically significant	OXM may be important for mediating some effects of OXY based on the worsened deterioration performance ratings for paroxetine alone Caveats: No PGx sub-analysis was conducted due to the small sample size in each group
***COMT* polymorphisms: Effect on opioid/oxycodone efficacy**				
Rakvag et al. (2005) [87]	207 Caucasian patients undergoing morphine treatment for cancer pain were selected and *COMT* Val/Met polymorphism was analyzed	PK measurements: Blood morphine, and metabolite exposure was recorded through a one-time blood collection test PD measurements: Pain was measured using the brief pain inventory questionnaire. Quality of life assessments, cognitive function, and functional status was also assessed PGx measurements: *COMT* Val/Met polymorphism at position 158 was probed	Patients with the Met/Met genotype received significantly lower morphine doses than the Val/Val genotype Serum concentrations of morphine and morphine glucuronides were significantly higher in Val/Met compared to Met/Met genotypes There was no difference between any genotype groups in the PD assessments	Val/Met polymorphism in *COMT* may be responsible for morphine PK; however, it does not seem to influence PD measurements
Laugsand et al. (2011) [85]	1579 cancer patients receiving opioids reported intensity of nausea and vomiting	PD measurements: Intensity of adverse drug reactions—nausea and vomiting PGx measurements: 96 SNPs in 16 candidate genes including *ABCB1*, *OPRM1*, and *COMT* were analyzed	Though three SNPs in the *COMT* gene were associated with the incidence of adverse drug reactions, none passed the Benjamini–Hochberg criterion for a 10% false discovery rate	*COMT* polymorphisms did not seem to affect opioid-mediated adverse reactions
***OPRM1* polymorphism: Effect on opioid/oxycodone efficacy**				
Lostch et al. (2009) [79]	Data were collected from 352 Caucasian patients with chronic pain treated with opioids for a minimum of 1 month	PD measurements: 24-h pain intensity and adverse drug reactions were recorded PGx measurements: Genotyping was carried out for *ABCB1*, *OPRM1*, *COMT*, and *CYP2D6*	*OPRM1**A118G* was the sole variant associated with 24-h pain scores; carriers of the *G* allele were more likely to experience pain *ABCB1**C3435T* significantly associated with opioid dosing No genetic associations were found for the incidence of adverse drug reactions	*OPRM1**G* allele carriers may have lower efficacy of opioids compared to homozygous *AA* carriers Caveats: Though a significant association was found, statistical significance failed in a subsequent ANOVA test
Zwisler et al. (2010) [80]	Oral OXY efficacy was tested in 33 healthy volunteers using pain tests	PD measurements: Pain threshold was tested using electrical stimulation and cold pressor tests; adverse events were recorded PGx measurements: *OPRM1* *(A118G)* and *ABCB1* (*C3435T* and *G2677T/A*) were determined	The *OPRM1* *A118G* variant was associated with reduced OXY efficacy Carriers of the *T* allele at the *C3435T* position had fewer side effects Carriers of the *T* allele at the *G2677T/A* polymorphism had enhanced response of OXY	*OPRM1* and *ABCB1* variants may contribute to a differential response to OXY
Zwisler et al. (2012) [86]	OXY-mediated analgesia was analyzed in 268 postoperative patients (OXY administered via IV)	PD measurements: Pain was rated by patients on a numerical rating scale; adverse reactions were monitored PGx measurements: *OPRM1* *(A118G)* and *ABCB1* (*C3435T* and *G2677T/A*) were assessed. The *CYP2D6* genotype was also analyzed	No association was found for pain intensity, incidence of adverse reactions, or OXY response between any genotypes Results did not change when patients were stratified according to their *CYP2D6* genotype-guided metabolizer status	*OPRM1* and *ABCB1* genotypes may not predict variable OXY response
Gong et al. (2013) [78]	Patients with cancer pain who were taking opioids (including OXY) were recruited	PD measurements: 24-h opioid usage was recorded and pain was assessed on a visual analog scale PGx measurements: *OPRM1* and *ABCB1* genotyping was carried out for the *A118G* and *C3435T* polymorphisms, respectively	Opioid doses required by the *OPRM1* *AG/GG* carriers were on average 54% higher than required for the *AA* carriers No differences in 24-h opioid usage were found between *ABCB1* genotypes	*OPRM1**A118G* polymorphism may be important for opioid activity
Cajanus et al. (2014) [81]	Association between the *OPRM1* *A118G* polymorphism and OXY-mediated analgesia was studied in 1000 women undergoing surgery for breast cancer (OXY administered via IV)	PD measurements: Cold and heat sensitivity was tested after surgery. The dose of IV OXY required to give pain relief was measured PGx measurements: Only the *OPRM1* *A118G* polymorphism was analyzed	*A118G* was significantly associated with the dose of OXY required for post-operative analgesia, with the *GG* phenotype requiring the highest and *AA* the lowest amount for pain relief *G* allele carriers also had higher post-operative pain ratings at baseline compared to the homozygous *AA* carriers	
Hwang et al. (2014) [82]	18 articles, totaling 4607 participants, were reviewed in a meta-analysis to determine whether there is an association between *OPRM1* *A118G* polymorphism and response to opioids	PD measurements: Primary analyses consisted of standardized mean difference measurements between opioid consumption in different variants of the *OPRM1* gene	*GG* carriers required significantly higher mean doses of opioid for pain relief compared to *AA* carriers	*OPRM1* polymorphism may be a determinant of opioid efficacy
Olesen et al. (2015) [74]	Meta-analyses from three previously published studies of Caucasian healthy volunteers to determine association with *OPRM1* variants and pain tests after OXY, morphine, and/or CR665 administration	Data from pain tests were extracted from original studies 6 polymorphisms were tested in *OPRM1* with others in the *OPRK1* and *OPRD1* genes	Different pain test responses (heat vs. visceral vs. muscle sensitivity) were associated with significantly different SNPs in the *OPRM1* gene The *A118G* polymorphism was weakly associated with reduced OXY analgesic response in the visceral pressure test	Choice of pain test in healthy volunteer studies can introduce confounding factors in comparing different studies
Ren et al. (2015) [83]	A systematic review–meta-analysis approach was applied to determine association between opioid efficacy and genetic polymorphisms	PD measurements: Analgesic efficacy, pain scores, and adverse drug reactions were picked as meta-analysis variables PGx measurements: Variants for *OPRM1*, *CYP3A4*, *CYP3A5*, and *ABCB1* were used to test association with opioid efficacy	*OPRM1**G* allele carriers at the *A118G* position consumed higher opioid doses for analgesia and had lower odds of adverse events *CYP3A4*1G* carriers consumed more opioids than **1/*1* carriers No associations were found with other genes	*OPRM1**A118G* polymorphism may be important for opioid efficacy
Khalil et al. (2017) [84]	153 postoperative pain patients receiving opioids were recruited	PD measurements: post-operative pain and opioid consumption were monitored PGx: *OPRM1* *(A118G)* and *COMT* (Val158Met and rs4633) were probed	*GG* allele carriers in the *OPRM1 A118G* position were associated with higher pain ratings following opioid treatment *AG/GG* carriers at *A118G* also had significantly higher opioid consumption	*OPRM1* may be important in variable response to opioids

Abbreviations: ABCB1: ATP-binding cassette sub-family B1; AUC: area under the curve; C_max_: maximum plasma concentrations; COMT: catechol O-methyl transferase; CYP2D6: cytochrome P450 2D6; CYP3A4: cytochrome P450 3A4; CYP3A5: cytochrome P450 3A5; DDI: drug–drug interaction; NM: normal/extensive metabolizer; norOXY: noroxycodone; norOXM; noroxymorphone; OPRD1: opioid receptor delta 1; OPRK1: opioid receptor kappa 1; OPRM1: mu-opioid receptor; OXM: oxymorphone; OXY: oxycodone; PD: pharmacodynamics; PK: pharmacokinetics; PGx: pharmacogenetics; PM: poor metabolizer; IV: intravenous.

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
