# Peer review of "Contribution of CYP2D6 Functional Activity to Oxycodone Efficacy in Pain Management: Genetic Polymorphisms, Phenoconversion, and Tissue-Selective Metabolism"

_pharmaceutics, 2021, doi:10.3390/pharmaceutics13091466_

Round 1

Reviewer 1 Report

  1. Table 1 - starting line 168 - Footnote should include definition of all abbreviations used in the Table so that the reader need not reference other uses of these abbreviations in the text of the manuscript.  This approach is substantially completed for Table 2.
  2. Line 460-2 - Authors should not claim an effect of voriconazole if the perceived change did not reach statistical significance.  It is unclear whether the inability to achieve statistical significance is a result of an underpowered study, or the natural variability that would occur if the study were repeated.  Best to say the study failed to demonstrate a significant effect of voriconazole.
  3.   Minor - I would prefer that paracetamol be replaced with acetaminophen, as my bias is that this is the more commonly used name for this analgesic drug.  Alternately, one of the two names could be placed in parentheses to completely inform the reader, regardless of their local appreciation for the generic name of this analgesic drug.

Author Response

  1. Table 1 - starting line 168 - Footnote should include definition of all abbreviations used in the Table so that the reader need not reference other uses of these abbreviations in the text of the manuscript.  This approach is substantially completed for Table 2.

We thank the reviewer for noticing this detail. A footnote to table 1 was added as suggested.

  1. Line 460-2 - Authors should not claim an effect of voriconazole if the perceived change did not reach statistical significance.  It is unclear whether the inability to achieve statistical significance is a result of an underpowered study, or the natural variability that would occur if the study were repeated.  Best to say the study failed to demonstrate a significant effect of voriconazole.

As per the reviewer’s comment, the sentence was modified as follows “Voriconazole coadministration did not significantly alter perceived subjective drug effect (VAS), or other observed behaviors”. (line 471)

  1. Minor - I would prefer that paracetamol be replaced with acetaminophen, as my bias is that this is the more commonly used name for this analgesic drug.  Alternately, one of the two names could be placed in parentheses to completely inform the reader, regardless of their local appreciation for the generic name of this analgesic drug.

We agree that acetaminophen is more commonly used, we indicated acetaminophen in the text and paracetamol in parenthesis in order to maintain the language used in the original paper (line 522).

Reviewer 2 Report

This manuscript presents a critical review on the role of CYP2D6 pharmacogenetics and phenotypes.

This is well written, with a final table resuming critically important papers in the field.

The review seem complete to me. There are some repetitions but this makes things clearer.

I find this review interesting and thus worth publishing particularly on a compound like oxydodone, which now has a very bad reputation for being addictive.

Author Response

  1. This manuscript presents a critical review on the role of CYP2D6 pharmacogenetics and phenotypes.
  2. This is well written, with a final table resuming critically important papers in the field.
  3. The review seems complete to me. There are some repetitions but this makes things clearer.
  4. I find this review interesting and thus worth publishing particularly on a compound like oxydodone, which now has a very bad reputation for being addictive.

We thank the reviewer taking time to review the manuscript and appreciate their positive and encouraging comments.

Reviewer 3 Report

This paper reviews the contribution of CYP2D6 to oxycodone efficacy in pain management focusing on the PK and PD changes of oxycodone by CYP2D6 polymorphisms. This topic is interesting, and the contents of this review is considerably fitted to the special issue topic. However, there are several points to be revised and my comments are as followings:

  1. Authors focused on the relationship between the PK especially in CYP2C6 mediated metabolism of oxycodone and PD mediated by µ-opioid receptor. This main issues seemed to be well chosen, but the structure is somewhat complicated. For example, some of PD parts are also mentioned the metabolism, which is overlapped to PK parts. Also in PK parts, CYP2D6 is only focused at first, but CYP3A4 and other phase II reaction are also important factors to determine PD effect. So, I recommend to re-organize the paper.
  2. The summarized table according to the sub-issues are required. Table 2  was not properly categorize the sub-issues.
  3. Some figures describing the relationships among  polymorphism of CYP2D6, PK and PD of oxycodone or oxymorphone are necessary.

Author Response

This paper reviews the contribution of CYP2D6 to oxycodone efficacy in pain management focusing on the PK and PD changes of oxycodone by CYP2D6 polymorphisms. This topic is interesting, and the contents of this review is considerably fitted to the special issue topic. However, there are several points to be revised and my comments are as followings:

  1. Authors focused on the relationship between the PK especially in CYP2C6 mediated metabolism of oxycodone and PD mediated by µ-opioid receptor. This main issues seemed to be well chosen, but the structure is somewhat complicated. For example, some of PD parts are also mentioned the metabolism, which is overlapped to PK parts. Also in PK parts, CYP2D6 is only focused at first, but CYP3A4 and other phase II reaction are also important factors to determine PD effect. So, I recommend to re-organize the paper.

We understand the point raised by the reviewer, however given the nature of the paper and to maintain the flow of the narrative, in some places an overlap of PK, PD, and PGx was inevitable. The subheadings have been modified to give a more precise meaning to the purpose of the paragraph, and, to make it easier for the reader to identify what is covered in the following section, tags were added to each title to clarify whether the section covers PK, PD, PGx, or a combination of the three.

  1. The summarized table according to the sub-issues are required. Table 2 was not properly categorize the sub-issues.

We have reorganized table #2. For instance, within the CYP2D6 category, studies are sub-categorized as “only PGx”, “only DDI”, or “PGx and DDI”, and studies are reorganized by publication date. In the COMT and OPRM1 sections, studies are ordered by date of publication. Appropriate abbreviations have been updated in the footnote.

  1. Some figures describing the relationships among polymorphism of CYP2D6, PK and PD of oxycodone or oxymorphone are necessary.

As suggested by the reviewer, a new Figure 2 (see below) was added to the manuscript to summarize the relationships between CYP2D6 polymorphisms and effects on PK and PD.

Figure 2 provides a summary of the relationship between oxycodone and oxymorphone pharmacokinetics, CYP2D6 genetic polymorphisms and pharmacodynamics.” (line 425)

Round 2

Reviewer 3 Report

I think it is acceptable.